# Fracture Resistance of 3D-Printed Occlusal Veneers Made from 3Y-TZP Zirconia

**DOI:** 10.3390/ma17092122

**Published:** 2024-04-30

**Authors:** Andreas Zenthöfer, Dennis Fien, Johannes Rossipal, Ali Ilani, Clemens Schmitt, Sebastian Hetzler, Peter Rammelsberg, Stefan Rues

**Affiliations:** Department of Prosthodontics, University of Heidelberg, 69120 Heidelberg, Germany; dennis.fien@icloud.com (D.F.); johannes.rossipal@med.uni-heidelberg.de (J.R.); sebastian.hetzler@med.uni-heidelberg.de (S.H.); peter.rammelsberg@med.uni-heidelberg.de (P.R.); stefan.rues@med.uni-heidelberg.de (S.R.)

**Keywords:** three-dimensional printing, zirconia, flexural strength, additive manufacturing, ceramics

## Abstract

The aim of this paper was to evaluate the fracture resistance of 3D-printed zirconia occlusal veneers (OVs) of different thicknesses and supported by different abutment materials. **Materials and Methods:** The standard OV of a natural molar was prepared and digitized using a laboratory 3D scanner. The resulting digital tooth abutment was milled either using cobalt–chromium (CoCr) or a fiber-reinforced composite (FRC). All the abutments were digitized and standardized OVs (30° tilt of all the cusps) designed with 0.4 mm, 0.6 mm, or 0.8 mm wall thicknesses. The OVs were fabricated using either the Programill PM7 milling device (Ivoclar Vivadent, PM) or one of two 3D zirconia printers, Cerafab 7500 (Lithoz, LC) or Zipro-D (AON, ZD). The ZD samples were only tested on CoCr abutments. The completed OVs were luted to their abutments and subjected to artificial aging, consisting of thermocycling and chewing simulation before fracture testing with a steel sphere (d = 8 mm) as an antagonist with three contact points on the occlusal OV surface. Besides the total fracture resistance F_u,tot_, the lowest contact force F_u,cont_ leading to the local fracture of a cusp was of interest. The possible effects of the factors fabrication approach, wall thickness, and abutment material were evaluated using ANOVA (α = 0.05; SPSS Ver.28). **Results:** The total fracture resistance/contact forces leading to failure ranged from F_u,tot_ = 416 ± 83 N/F_u,cont_ = 140 ± 22 N for the 0.4 mm OVs fabricated using LC placed on the FRC abutments to F_u,tot_ = 3309 ± 394 N (ZD)/F_u,cont_ = 1206 ± 184 N (PM) for the 0.8 mm thick OVs on the CoCr abutments. All the factors (the fabrication approach, abutment material, and OV wall thickness) had an independent effect on F_u,tot_ as well as F_u,cont_ (*p* < 0.032). In pairwise comparisons for F_u,tot_ of the OVs luted to the CoCr abutments, the ZD samples statistically outperformed the LC- and PM-fabricated teeth irrespective of the thickness (*p* < 0.001). **Conclusions:** Within the limitations of this study, the printed occlusal veneers exhibited comparable fracture resistances to those of the milled variants. However, more resilient abutments (FRC as a simulation of dentine) as well as a thinner wall thickness led to reduced OV fracture resistance, suggesting that 0.4 mm thick zirconia OVs should not be unreservedly used in every clinical situation.

## 1. Introduction

Zirconia is becoming increasingly important in restorative dentistry [1,2,3]. With modifications such as the sintering temperature and the adjustment of the alumina and/or yttria content, the mechanical as well as esthetic parameters of a specific zirconia material can be varied, thereby enabling the clinical use of monolithic zirconia restorations. However, improved esthetical properties towards the level of lithium disilicate ceramics exist, in general, alongside a reduction in fracture strength [3,4]. 

When it comes to the treatment of worn dentition due to attrition, abrasion, or erosion, frequently, the treatment mandate is to restore occlusal function and esthetics, and therefore the patient’s oral health-related quality of life without extended invasiveness [5,6]. However, the reconstruction of occlusal dimensions should also prevent possible sequelae such as TMD [6,7]. In particular, OVs made of lithium disilicate ceramics are used and show adequate clinical outcome [8,9]. As zirconia shows—amongst its other favorable material properties—higher flexural strength, higher Young’s modulus, and less antagonistic abrasion [4,10] compared with lithium disilicate ceramics, it could also be an interesting material for OVs used for occlusal reconstruction. Primarily, this can be helpful to treat patients with parafunctional behavior, and therefore higher chewing forces. Nonetheless, the major drawback of zirconia is that it requires a more sensitive luting process compared to glass ceramics, which could lead to more frequent debonding [11,12]. 

Almost all zirconia restorations are fabricated following the CAD/CAM approach with a computer aided design and subtractive computer aided milling. Recently, additive manufacturing via the 3D printing of so-called green bodies has been introduced to the market. These 3D printers selectively light-cure a slurry containing of an acrylic binding system and zirconia particles layer by layer using stereolithography (LCM) wavelengths [13,14,15]. 

The completed objects are cleaned afterwards, subjected to the debinding of the resin content, and then finally sintered [13,14,15]. 

The previous studies on the material properties of dental specimens show a lower, but sufficient fracture strength compared to their milled counterparts [14,15,16]. These studies showed that more voids and flaws can be found in the material itself or in the layer interface areas created during the printing process in comparison to an industrially prefabricated zirconia blank [14,15,16].

Other previous studies have also shown that these voids and flaws are caused by the necessary post-processing of specimens such as cleaning using isopropanol and debinding and sintering procedures [17].

However, one strength of 3D zirconia printing is the opportunity of manufacturing thinner and geometrically demanding restorations as objects are not subjected to tooling stress and milling radius correction [13,18]. In this interesting approach, the design freedom may be especially favorable in the fabrication of (occlusal) veneers. 

A previous study found that anterior veneers fit the inner and marginal parts almost as well as milled veneers [19]. The cross-sectional views also showed that the internal width of the cement gap was homogenous, which makes them much more promising than the milled variant requiring radius correction [19]. Beyond these observations, the literature on the comparison of 3D-printed and milled veneers is missing.

However, as these previous devices have not been clinically certified, this technique seems to have come to a dead end. Recently, a 3D printer with CE, EPA, and ISO certifications has become available, which refreshes the idea of LCM’s potential. A previous study revealed sufficient fracture strength for this printed zirconia and promising reliability in biaxial strength testing [15]. However, it remains unclear if the fracture resistance of real restorations with more complex geometries than the samples used in biaxial strength tests is comparable for milled and 3D-printed OVs. Furthermore, the fracture resistance of OVs will be influenced by the cement gap width, and thus by the fit of the OVs.

Working hypotheses: The forces associated with OV fracture increase with an increasing wall thickness (1), milled OVs can withstand higher forces than 3D-printed OVs can (2), and the underlying abutment material affects the fracture resistance (3).

## 2. Methods

The complete study workflow with the 120 OV test samples differing in abutment material, OV fabrication method, and OV wall thickness is summarized in Figure 1.

### 2.1. Abutment Fabrication

A natural molar tooth (tooth 46) served as the basis for this investigation on OVs and was embedded in acrylic resin (Technovit 4071, Kulzer, Wehrheim, Germany) such that the tooth axis was oriented vertically. The tooth was then fixed with a tilt of 30° and a predefined rotation around its axis for each cusp using a parallel milling device with height control. This allowed for the planar preparation of the cusps with a 30° tilt with regard to the horizontal plane and controlled material removal (about 0.5–0.6 mm) using diamond-coated burs. For circumferential preparation along the finishing line, the tooth was fixed vertically in the parallel milling device. The rest of the tooth surface was prepared without help of the paralleling device. The final prepared tooth was digitized with a laboratory scanner (D2000, 3Shape, Copenhagen, Denmark), and the abutment’s exact cusp orientation (30° tilt) and the minimum radii along the cusps´ edges were optimized (Geomagic Design X, 3D Systems, Rock Hill, SC, USA). This resulted in data on the digital reference geometry of a prepared molar abutment tooth.

Based on these digital data, the abutment tooth was replicated 72 times using cobalt–chromium alloy (CoCr; Colado, Ivoclar Vivadent AG, Schaan, Liechtenstein) and 48 times using a fiber-reinforced composite (FRC; Trinia, Bicon, Boston, MA, USA) with a 5-axis milling machine (PM7, Ivoclar Vivadent). Finally, all the abutment teeth were sandblasted (50 µm alumina particles, 0.1 MPa).

### 2.2. OV Design and Fabrication

Individual scans (D2000, 3Shape) of each replicated abutment tooth served as the basis for the OV design (Dental Designer, 3Shape) with a constant wall thickness, w. Depending on the test group, one of three wall thicknesses (w = 0.4 mm, w = 0.6 mm, or w = 0.8 mm) was chosen. The marginal fit during the design was 20 µm, whereas the internal gap was 60 µm wide (Figure 2). Wall thickness of w = 0.6 mm was selected as minimum clinical thickness for milled tetragonal zirconia polycrystal doted with 3 mol% Y_2_O_3_ (3Y-TZP); w = 0.4 mm and w = 0.8 mm were chosen to control the impact of thinner and thicker restorations.

For each test group differing in abutment material and wall thickness, *n* = 8 OVs were either milled from zirconia blanks (IPS e.max ZirCAD LT, Ivoclar Vivadent) using a 5-axis milling machine (PM; PM7, Ivoclar Vivadent) or the Cerafab 1600 zirconia printer (LC; slurry: LithaCon 3Y 210, Lithoz, Vienna, Austria). Another *n* = 8 OVs were printed with a second 3D printing system (ZD; printer: Zipro-D, slurry: Inni Cera BCM 1000, AON, Seoul, South Korea) only for the test groups associated with the CoCr abutments. The LC samples were nested vertically, whereas the ZD samples were nested horizontally. Supports on the occlusal surface of the ZD veneers were omitted in the regions that came into contact later with the antagonist. All the fabrication and cleaning procedures followed the manufacturers’ instructions. The PM OVs were sintered (Programat S1, Ivoclar Vivadent) for 9.5 h at a final temperature of 1500 °C. After cleaning the LC samples with the recommended solvent (LithaSol 30, Lithoz), they were fired (debinding and sintering) for 50 h at temperatures up to 1450 °C (HTCT 08/17, Nabertherm, Lilienthal, Germany) as described in the previous studies. The completed ZD veneers underwent airbrush cleaning with isopropanol of 99.5% purity before they were fired (debinding/presintering) at up to 1100 °C for 30 h (ZIRFUR, AON), and then at 1500 °C for 5 h (HTCT 08/17, Nabertherm). Also, see this reference [15] for details of the post-processing of the printed parts. An overview of the materials used to produce the OVs as well as the artificial abutment teeth is given in Table 1.

### 2.3. Sample Handling and Aging

The inner surfaces of all the zirconia OVs were sandblasted (50 µm alumina particles, 0.1 MPa) and conditioned using an MDP-containing primer (Clearfil Ceramic Primer Plus; Kuraray, Hattingen, Germany). As described above, the abutment teeth were sandblasted before the scanning process. The FRC samples were conditioned with the recommended primer (Cera Resin Bond 1 + 2, Shofu, Tokyo, Japan), whereas for the CoCr abutments, no primer was applied. The adhesive cementation (Panavia 21, Kuraray) of the OVs was carried out using a universal testing device (Z005, Zwick/Roell, Ulm, Germany) with a vertical and centric force of 200 N, which was kept constant for 6 min. Before cementation, the abutments were stored at 42 °C such that the temperature during cementation was about that of a body. After cementation, all the specimens were stored under humid conditions (100% humidity) at 37 °C for 24 h before being placed in 37 °C warm deionized water for 30 d ± 2 d. During the 30 d of water storage, further artificial aging took place, consisting of 10,000 thermal cycles between 6.5 °C and 60 °C (TC1, SD Mechatronik, Feldkirchen, Germany) and 1,200,000 chewing cycles with a force magnitude of F = 108 N. Load application during chewing simulation and fracture testing was identical. After aging, the OVs were checked for damage with a digital microscope (Smartzoom 5, Zeiss, Jena, Germany).

### 2.4. Fracture Tests

The aged OVs were loaded with an 8 mm steel ball and a crosshead speed of 0.5 mm/min (Z005, Zwick/Roell). The sample was placed on a horizontally oriented ball bearing (Figure 3), thus ensuring a purely axial load (F_res_) application with its line of action intersecting the sphere center (SC). With the chosen OV design, 3 contact points (CP_1/2/3_) formed between the antagonist and OV (Figure 4a). Since there was relative movement between the steel ball and OV at the contact points during testing, it could be assumed that the maximum possible friction forces (µ: coefficient of friction) were acting, i.e., each contact force F_i_ was tilted by arctan µ with respect to the normal vector on the respective cusp surface (Figure 4b). Since all the forces met at the reduction point (RP), the balance of moments was automatically fulfilled, resulting in 3 equations (balance of forces) for the 3 unknown contact forces F_cont,i_. Since the calculated contact forces decreased with an increasing coefficient µ, a rather high value µ = 0.2 between zirconia and steel was assumed (for conical crown retention with polished surfaces, µ = 0.15 was given for the combination CoCr/ZrO_2_ with polished surfaces [20]), resulting in the contact forces F_cont,1_ = 0.424 ∙ F_res_; CP_cont,2_ = 0.310 ∙ F_res_; and CP_cont,3_ = 0.312 ∙ F_res_. During the fracture tests, video surveillance was installed, and the structure-borne sounds were recorded, signaling the cracking of the OVs. As soon as a sound indicated that a crack occurred, the test was stopped, and the sample was inspected (Smartzoom 5, Zeiss) for possible damage. In case of crack formation, the crack position(s) were noted, and the test force at this time was defined as the total fracture resistance F_u,tot_. Furthermore, the minimal contact force acting on a cracked cusp was defined as the maximum contact force F_u,cont_. If no damage was identified, the fracture testing of this sample continued with the methods described above.

It should be noted that with linear elastic materials, the complete stress state in the sample can be described as the linear superposition of 3 separate load cases, each with F_cont,i_ acting on the respective cusp. Based on a published paper on the FE analysis of crowns loaded on one cusp [21], it was seen that considerable stresses only formed in the loaded cusp, while the stresses that occurred in the other (unloaded) cusps were negligible. Consequently, fractures originating from different loading sites were directly correlated to the respective contact forces for OVs.

### 2.5. Statistical Analysis

Statistical analyses were performed using SPSS Ver. 28 (IBM; New York, NY, USA). The mean values and standard deviations (SDs) for all the groups were computed and visualized using boxplot diagrams. QQ plots were inspected and Shapiro–Wilk tests were performed to prove normality of our data. The impact of different factors (abutment material, fabrication approach, and wall thicknesses) on F_u,tot_ and F_u,cont_ was evaluated separately with ANOVA and post hoc Tukey tests. As higher standard deviations were seen for LC—in accordance with previous studies—Welch ANOVA was conducted to address this statistical demand. The local statistical significance α = 0.05 was assumed. 

## 3. Results

All the samples passed the aging simulation without visible damage or decementation. In the fracture tests, crack detection worked without problems in 72% of the samples, i.e., visible cracks were present on the OV surface after ending one fracture test as soon as a sound signal was recorded, indicating a fracture. With a few exceptions, the OVs for which the fracture test had to be restarted had CoCr abutments. In most cases (24% of the samples), a visible fracture was detected after a second test. Three or even four repetitions were only necessary for 4% of the samples.

The mean total fracture resistances F_u,tot_ and mean contact forces F_u,cont_ causing fractures are summarized in Table 2 and visualized in Figure 5a,b. The total fracture resistance F_u,tot_ ranged from 416 N ± 83 N for the 0.4 mm thick LC OVs placed on the FRC abutments to 3309 N ± 394 N for the 0.8 mm thick ZD variants luted to the CoCr abutments. The fabrication approach (PM, LC, or ZD), abutment material (CoCr or FRC), and wall thickness of the OVs (0.4 mm, 0.6 mm, or 0.8 mm) independently affected the F_u,tot_ (*p* < 0.001). The OVs with a 0.8 mm wall thickness had about three times more fracture resistance than the respective OVs with a 0.4 mm wall thickness did, which is slightly less than factor four and is related to bending-dominated problems and the doubling of the wall thickness. All the pairwise comparisons of the groups differing in wall thickness were significant (*p* < 0.001). A similar finding was given for F_u,cont_, with all the factors significantly affecting the local fracture force (*p* < 0.032). Here, it is shown that the 0.4 mm thick OVs (MP and LC) formed cracks on the cusps exposed to mean contact forces of 172 N and 140 N, respectively. The highest mean contact force for the milled OVs placed on the CoCr abutments was 1206 N. The F_u,cont_ values increased with an increasing wall thickness for the test groups, and all the pairwise test data were highly significant (*p* < 0.001).

Regarding the fabrication approach, the printed OVs made using LC or the milled OVs showed comparable F_u,tot_ values (*p* = 0.941), whilst the fracture resistance of the ZD OVs was about 20% higher (*p* < 0.001) when only regarding the OVs placed on the CoCr abutment teeth. With the FRC abutment teeth, again, the fracture resistance F_u,tot_ of the PM and LC test groups was comparable (*p* = 0.677). When analyzing the maximum contact forces the OVs could withstand, the only significant difference with regard to the fabrication method was found between the LC- and ZD-fabricated OVs placed on the CoCr abutments (*p* = 0.041).

As given in the Methods section, the distribution of the resulting test force among the three contact points was calculated based on the OV geometry. The forces at the contact points CP_2_ and CP_3_ were almost identical, but about 25% smaller than the contact force acting at CP_1_. Table 3 shows the number of fractures originating from one or more of the three different contact points. As was expected, 75.0% of the OVs had cracks in the vicinity of CP_1_, whereas the OVs showed cracks next to CP_2_ and CP_3_ in 56.7% and 45.8% of the cases, respectively. For every fabrication approach, an exemplary image of a fractured OV is presented in Figure 6. In most cases, the cracks formed around the contact points (circular segments). Only for a few cases, linear cracks through the center of the contact point areas were identified.

## 4. Discussion

The first study hypothesis that the wall thickness affects the fracture resistance was confirmed. The second hypothesis that the milled OVs are superior compared to the 3D-printed OVs was rejected. The last hypothesis that the underlying abutment materials had an effect was accepted.

Regarding the impact of the fabrication method, the 3D-printed OVs did not perform worse than their milled counterparts. Here, one might balance the favorable effects of design freedom against the possible voids and flaws, which can cause cracks and lead to failure, as seen for 3D-printed zirconia in the previous studies [14,15], thus lowering the fracture resistance. However, the fitting and homogenous layering of luting cement plays a crucial role. With an increasing cement layer thickness, the resilience increases. The fracture resistance of a thin-walled restoration supported by such a cement layer will therefore decrease with an increasing cement layer width. This effect has already been demonstrated in a test with layered specimens [22]. For dental restorations, this means that not only does the zirconia´s fracture strength affect the fracture resistance, but the fit and final position of the restoration after cementation can also have tremendous impact on load bearing capacity. In previous studies, it has been demonstrated for anterior veneers that printing in particular enables a good internal fit when it comes to specific geometries as no bur diameters or radii have to be considered with printing [19,23,24]. For OVs, no studies on the internal and marginal fits are available yet, and this was not conducted in this study. Fit of all, the OVs in this investigation were excellent, and manual adjustment was omitted. However, with a planned cement gap width of 60 µm, already small differences between the test groups might have resulted in considerable differences. The fit of the printed and milled OVs was not measured here and should be the topic of future research. Nonetheless, one might accept that the 3D-printed restorations in this study had a good fit, and therefore, thin and homogenous cement layering. In order to standardize the possible effects, for all the fabrication strategies, including milling, the same fitting parameters were used, and luting was performed under standardized conditions (a constant pressure and curing time) as well. In this context, one might mention that the milled variant should not be portrayed as being inferior. The scanned preparation was digitally post-processed, i.e., the sharp edges were removed from the digital abutment typodont such that no radius corrections were necessary during the design of the milled OVs.

Unsurprisingly, regarding the underlying abutment material, a significant effect was found. The FRC abutments rather resemble a tooth made solely of dentin; FRC is more resilient than enamel. On the other hand, the CoCr alloy is 2–3 times as stiff as enamel. In clinical cases, the abutment teeth prepared for thin-walled OVs will likely still have an enamel layer left above the dentin; thus, the mechanical behavior of a real abutment ranks between the two tested abutment tooth samples in this study. The thinnest OVs in this study could withstand mean resultant forces between about 400 N and 500 N on the FRC abutments and 900 N and 1400 N on the CoCr abutments. With 500 N being the typical threshold for clinical recommendation in the posterior region, it could be concluded that even 0.4 mm thick zirconia veneers are suitable for clinical use. However, when looking at the contact forces F_u,cont_ correlating with crack formation (Figure 5b), it becomes clear that the 0.4 mm thick zirconia OVs will likely be damaged in vivo if a force between 200 N and 300 N acts on a single cusp.

As already stated in the Results section, the fracture resistance F_u_ of the samples exposed predominantly to bending (e.g., long beams) will increase with the squared beam height h, i.e., F_u_~h^2^. Using this simple analogy, the fracture resistance of OVs with 0.4 mm thick or 0.8 mm thick walls should show a ratio of 1:4. However, with a given material strength, the deflection of a beam at the fracture u_u_ decreases: F_u_~1/h. This means that a thin-walled OV will show a comparatively large deflection beneath the loading site. This deflection is impeded by the underlying materials, with the effect being most dominant for the thinnest OVs. This might explain that we found only three-times-higher fracture forces when the wall thickness was doubled. For the predominant fracture mode, concentric crack formation around the contact points was observed. In this case, it is likely that the fracture origin was located on the occlusal surface since high tensile stresses are present around the contact points. On the other hand, linear cracks through the contact point centers were probably correlated with the fracture origins located on the inner crown surface. The latter fracture mode would be dominant if the fit of the OVs was bad, i.e., for large cement gaps enabling rather large deflections. To definitely clarify the fracture origins, fractography based on SEM images of the fracture surfaces could be included in future studies.

A comparison with other studies is often difficult since all the above mentioned parameters (material strength, material stiffnesses, geometry, and the fit of the OV) as well as the loading conditions influence fracture resistance. Furthermore, the flaws created during material processing may differ between different research groups. When looking at the clinical reality, the majority of patients in need of long-term occlusal reconstruction are treated with OVs fabricated using milled or pressed lithium disilicate ceramics [5,8]. While the luting of these restorations is reliable, the flexural strength is roughly half of that of zirconia. In laboratory studies on thinner occlusal OVs made of lithium disilicate, a difference in the load needed to cause a fracture was also seen depending on the underlying substrate (dentine vs. enamel) [25]. The same was true for the variation in thicknesses [26,27]. However, while the thematic clinical studies included a manageable number of cases, the long-term performance of regularly thick lithium disilicate restorations seems to be good [5,8]. Schlichting et al. [9] found that even ultrathin OVs seem to be promising if the preparation includes enamel. However, a recent report by Czechowski et al. [26] showed that the fracture resistance for zirconia restorations is substantially higher than those made of lithium disilicate. Thus, thinner OVs can be made if zirconia is used. To this end, the luting of zirconia is still challenging and requires a structured technique-sensitive protocol. As zirconia has no glass phase, sandblasting or tribochemical conditioning in combination with MDP-containing primers and / or adhesive cements—as used in this study—have yielded the most reliable bond with the tooth substrates [11,12]. It has been demonstrated that a reliable bond was also achieved for OVs made from zirconia [28].

### Strengths and Limitations

To the knowledge of the authors, this is the first systematic study on the loading fracture of real dental restorations made using the new printing technology, and a series of standardizations were applied in order to make a reliable statement on the fracture resistance of OVs. The potential of 3D-printed zirconia OVs in the posterior dentition is high as design freedom, minimal invasiveness, and high flexural strengths of the used restoration material are especially needed in the adjustment of occlusion (i.e., worn dentition, amelogenesis imperfecta). However, for its use in anterior veneers, esthetical improvements such as translucency are desirable. One limitation of this study is that the ZD groups were only tested on the CoCr abutments, which restricts the interpretation of how those OVs would perform on a more flexible substrate. Future studies are necessary if the performance of ZD OVs on FRC abutment teeth is similar to the findings for PM- and LC-fabricated OVs. One should also keep in mind the rather small sample size per group of *n* = 8. However, the group differences were very large, and an a priori power calculation suggested that it would be redundant to consider a larger sample size. With the actual sample size, the statistical effects proving or disproving the working hypotheses could be shown. With a larger sample size, it might have been possible to show more significant differences in pairwise post hoc tests. The fit of the OVs was only qualitatively controlled in this investigation. The fracture forces, as already mentioned above, decreased with an increasing cement gap width and the use of a resilient, resin-based adhesive. Consequently, the findings in this study could have been influenced to some extent by the differences in fit between the test groups. Again, since the fit was not quantitatively assessed in this investigation, further studies are necessary to clarify this issue.

## 5. Conclusions

Within the limitations, the printed occlusal veneers exhibited comparable fracture resistances to those of the milled variant. However, more flexible abutments (FRC as a simulation of dentine) with thinner walls have reduced fracture resistances, suggesting that the 0.4 mm thick zirconia OVs should not be unreservedly used in every clinical situation. In particular, the ZD-printed OVs seem to be interesting as they are allowed for clinical use. Here, further studies are needed to determine the fit properties.

## Figures and Tables

**Figure 1 materials-17-02122-f001:**
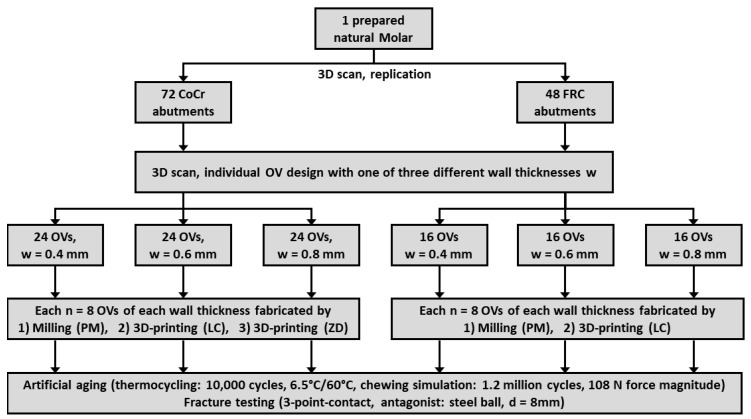
Study design and test groups of this investigation.

**Figure 2 materials-17-02122-f002:**
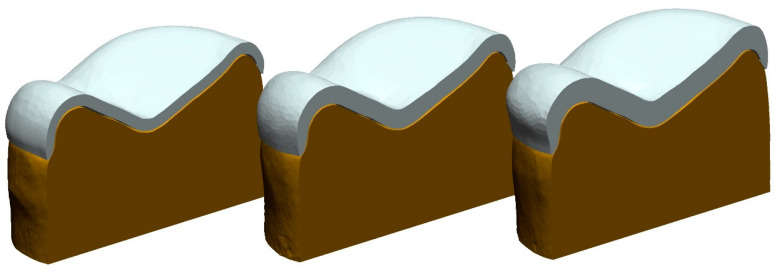
Exemplary OV designs for wall thicknesses w = 0.4 mm (**left**), w = 0.6 mm (**center**), and w = 0.8 mm (**right**). Cement gap width was 20 µm at the margin and 60 µm along the interior surface.

**Figure 3 materials-17-02122-f003:**
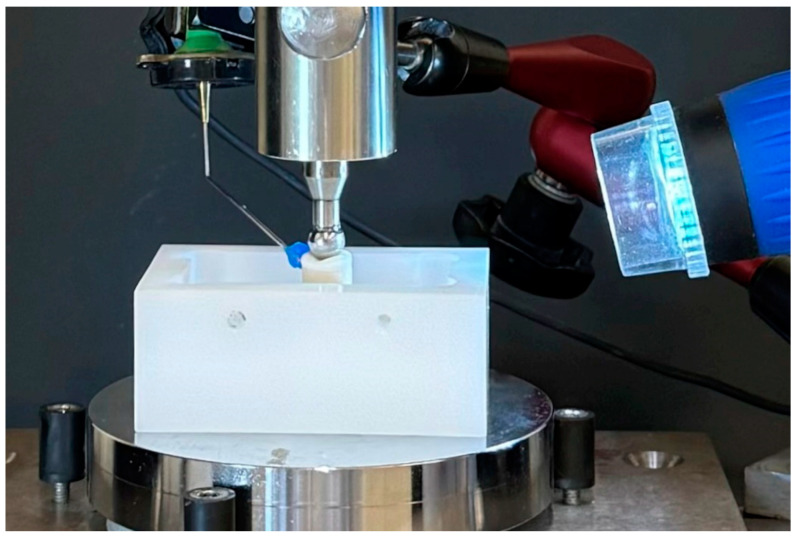
Fracture test setup with steel ball as antagonist. To exclude transverse forces, samples were placed on a ball bearing. Simultaneous video and structure-borne sound signal recordings served to identify cracking of the OVs.

**Figure 4 materials-17-02122-f004:**
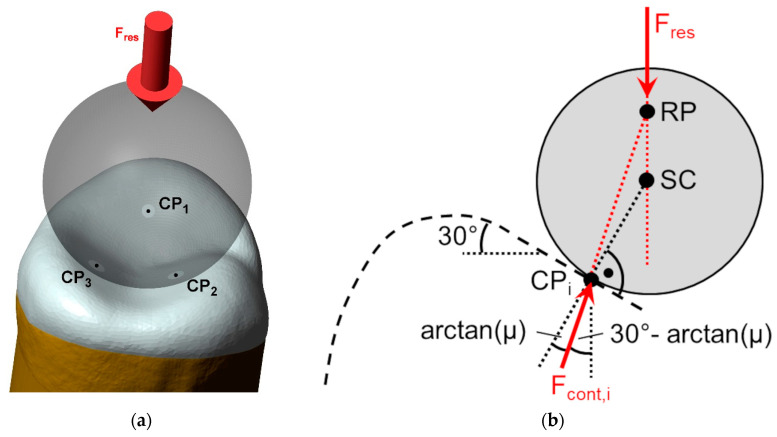
Three contact points (CP_1_, CP_2_, and CP_3_) formed during load application F_res_ on the OV via the steel ball (**a**). Under assumption that the steel ball slid (µ: coefficient of friction) downwards with increasing load F_res_, each contact force was tilted by 30° arctan (µ) with respect to the vertical axis (**b**).

**Figure 5 materials-17-02122-f005:**
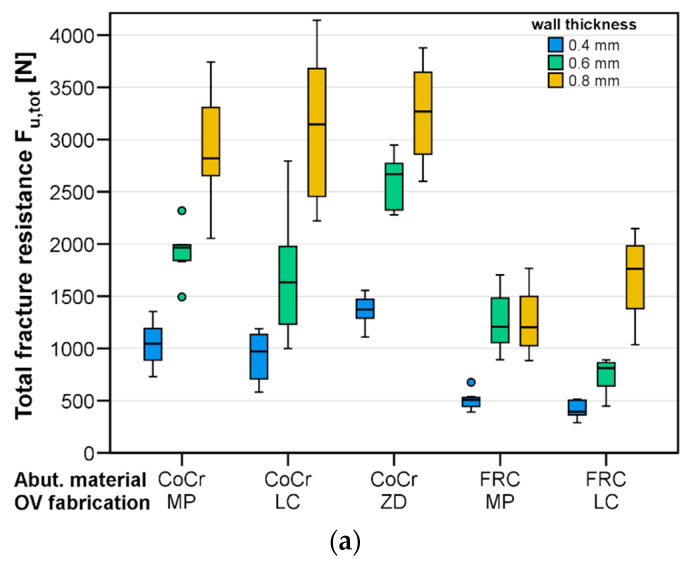
Boxplot diagrams showing the results for the total fracture resistance (**a**) and the maximal contact force the samples could withstand (**b**). The highest fracture resistances were seen for 0.8 mm thick ZD OVs on CoCr abutment teeth, the lowest for 0.4 mm thick LC OVs on FRC abutment teeth.

**Figure 6 materials-17-02122-f006:**
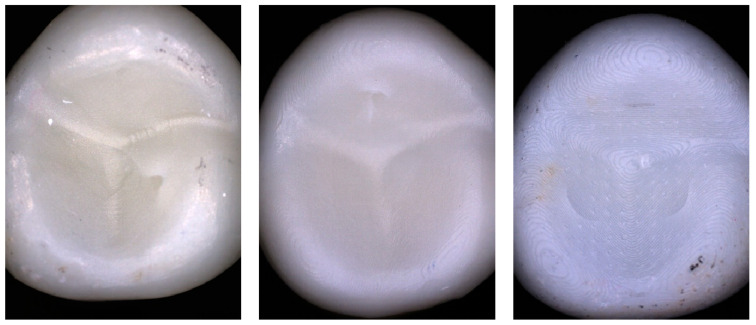
Exemplary images of fractures samples: a milled OV (PM) with a crack originating from CP_2_ (**left**), a 3D-printed OV (LC) with a crack originating from CP_1_ (**center**), and a 3D-printed OV (ZD) with cracks originating from CP_2_ and CP_3_ (**right**)_._

**Table 1 materials-17-02122-t001:** Materials used for the occlusal veneers as well as artificial abutment teeth fabrication.

	Occlusal Veneers			Abutment Teeth	
**Abbreviation**	**PM**	**LC**	**ZD**	**CoCr**	**FRC**
**Fabrication method**	Milling	3D Printing		Milling	
**Material**	3Y-TZP	3Y-TZP	3Y-TZP	CoCr alloy	FRC
	IPS e.max	LithaCon	Inni Cera	Colado CoCr4,	Trinia,
	ZirCAD LT,	3Y 210,	BCM 1000,		
	Ivoclar Vivadent	Lithoz	AON	Ivoclar Vivadent	Bicon
**Machine**	PrograMill PM7	Cerafab 1600	Zipro-D	PrograMill PM7	
**Firing**	9.5 h, 1500 °C Programat S1, Ivoclar Vivadent	50 h, 1450 °C HTCT 08/17, Nabertherm	30 h, 1100 °C ZIRFUR, AON + 5 h, 1500 °C HTCT 08/17, Nabertherm	-	-

**Table 2 materials-17-02122-t002:** Fracture test results for the test groups differing in abutment material, fabrication method, and OV wall thickness.

Abutment Material	Fabrication Method	Wall Thickness [mm]	F_u,tot_ [N]		F_u,cont_ [N]	
			Mean	SD	Mean	SD
CoCr	PM (milling)	0.4	1042	208	375	114
		0.6	1917	229	715	162
		0.8	3046	395	1206	184
	LC (3D printing)	0.4	924	234	339	93
		0.6	1868	273	634	124
		0.8	3302	514	1079	223
	ZD (3D printing)	0.4	1367	143	452	92
		0.6	2565	240	810	80
		0.8	3309	394	1146	289
FRC	PM (milling)	0.4	504	86	172	34
		0.6	1190	300	375	94
		0.8	1272	318	413	92
	LC (3D printing)	0.4	416	83	140	22
		0.6	779	112	255	50
		0.8	1680	394	570	139

**Table 3 materials-17-02122-t003:** Number of fractures occurring on each of the three cusps of the OVs.

	Cracks Originating from Contact Point(s)						
	CP_1_	CP_2_	CP_3_	CP_1_ and CP_2_	CP_1_ and CP_3_	CP_2_ and CP_3_	all CPs
*n* [-]	31	14	8	20	13	8	26
*n*/Σ*n*_i_ [%]	25.8	11.7	6.7	16.7	10.8	6.7	21.7

## Data Availability

The source data can be made available upon request.

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
