# Peer review of "Fracture Resistance of 3D-Printed Occlusal Veneers Made from 3Y-TZP Zirconia"

_materials, 2024, doi:10.3390/ma17092122_

Round 1

Reviewer 1 Report

Comments and Suggestions for Authors

This manuscript report the 3D-Printed Occlusal Veneers Made from 3Y-TZP Zirconia and their characterizations. The concepts and results of this work are fine so that I recommend this manuscript to be published in Materials journal.

Reviewer 2 Report

Comments and Suggestions for Authors

This study is interesting because the materials and systems are new.

The product descriptions are difficult for readers to understand, so I think it would be easier to understand if there was a table or something like that.

The experimental results are reasonable.

I think it would be nice if the explanation of Figure 6 was more detailed.

In the future, I think it would be interesting to conduct SEM observation of the fractured surface in the next test.

Reviewer 3 Report

Comments and Suggestions for Authors

Dear authors,

Thank you for submitting your manuscript, "Fracture Resistance of 3D-Printed Occlusal Veneers Made from 3Y-TZP Zirconia," for consideration in Materials.

Overall, the study's aim of comparing the fracture resistance of 3D-printed and milled OVs made from zirconia was adequately addressed in the conclusion. The results support the main findings that printed OVs exhibit comparable fracture resistances to milled variants, but more flexible abutments and thinner wall thicknesses lead to reduced fracture resistance.

After reviewing your manuscript, I would like to provide my feedback.

1. Perform a thorough language edit to improve the clarity and readability of the manuscript.

2. Consider discussing the limitations of the small sample size and its potential impact on the results.

3. Acknowledge the restriction of testing ZD groups only on CoCr abutments and suggest future studies to investigate their performance on more flexible substrates.

4. Mention the lack of OV fit measurements as a limitation and recommend future research to address this aspect.

In summary, the study is well-designed and contributes to the growing body of knowledge on 3D-printed dental restorations. With the suggested revisions, the manuscript has the potential to be a valuable addition to the literature in this field.

Sincerely,

Comments on the Quality of English Language

Moderate editing of the English language required

Reviewer 4 Report

Comments and Suggestions for Authors

This manuscript presents a well-conducted study on the fracture resistance of 3D-printed and milled zirconia occlusal veneers, using different abutment materials and varying wall thicknesses. The experimental design is robust, the methodologies are well explained, and the results are clearly presented. The findings are relevant to the field of restorative dentistry, providing valuable insights into the performance of enhanced manufacturing techniques for dental prostheses. However, some improvements are in need.

Major Comments:

1.      The introduction effectively sets up the study by discussing the relevance of zirconia in restorative dentistry and the evolution of fabrication methods. However, it would be great to include a deeper analysis of previous studies comparing 3D-printed and milled veneers, specifically those that highlight the mechanical properties and clinical outcomes.

2.      The optimization of the specific thicknesses for the veneers could be better justified. How do these thicknesses relate to clinical practices or previous research findings?

3.      For statistical analysis, samples numbers should be provided. The Normality and Lognormality Tests should be run before ANOVAs tests.

4.      For discussion part, it could be strengthened by discussing potential clinical implications of the findings in more detail, such as the suitability of different veneer types for specific patient groups or clinical conditions.

5.      For Figures and tables legend, more detailed descriptive statistics should be provided.

Comments on the Quality of English Language

Minor grammatical corrections are needed.
